# Physics-Regularized Multi-Modal Image Assimilation for Brain Tumor Localization

**Michal Balcerak**[1][*], **Tamaz Amiranashvili**[1,2], **Andreas Wagner**[2],

**Jonas Weidner**[2], **Petr Karnakov**[3], **Johannes C. Paetzold**[4], **Ivan Ezhov**[2],

**Petros Koumoutsakos**[3], **Benedikt Wiestler**[2][†], **Bjoern Menze**[1][†]

[1]University of Zurich, [2]Technical University of Munich
[3]Harvard University, [4]Imperial College London

## Abstract

Physical models in the form of partial differential equations serve as important priors for many under-constrained problems. One such application is tumor treatment planning, which relies on accurately estimating the spatial distribution of tumor cells within a patient's anatomy. While medical imaging can detect the bulk of a tumor, it cannot capture the full extent of its spread, as low-concentration tumor cells often remain undetectable, particularly in glioblastoma, the most common primary brain tumor. Machine learning approaches struggle to estimate the complete tumor cell distribution due to a lack of appropriate training data. Consequently, most existing methods rely on physics-based simulations to generate anatomically and physiologically plausible estimations. However, these approaches face challenges with complex and unknown initial conditions and are constrained by overly rigid physical models. In this work, we introduce a novel method that integrates data-driven and physics-based cost functions, akin to Physics-Informed Neural Networks (PINNs). However, our approach parametrizes the solution directly on a dynamic discrete mesh, allowing for the effective modeling of complex biomechanical behaviors. Specifically, we propose a unique discretization scheme that quantifies how well the learned spatiotemporal distributions of tumor and brain tissues adhere to their respective growth and elasticity equations. This quantification acts as a regularization term, offering greater flexibility and improved integration of patient data compared to existing models. We demonstrate enhanced coverage of tumor recurrence areas using real-world data from a patient cohort, highlighting the potential of our method to improve model-driven treatment planning for glioblastoma in clinical practice.

## 1 Introduction

The management of gliomas, particularly glioblastomas, is highly challenging due to their infiltration beyond the tumor margins detectable in medical imaging. This infiltrative behavior complicates the precise personalization of radiotherapy. Current clinical practice applies uniform radiation to a 1.5 cm margin around the active tumor core visible in Magnetic Resonance Imaging (MRI) scans. However, this approach overlooks critical factors such as heterogeneous infiltration patterns, varying brain tissues, and anatomical barriers, failing to account for the highly individualized dynamics of

---

[*]Corresponding author: michal.balcerak@uzh.ch
[†]Contributed equally as senior authors.

38th Conference on Neural Information Processing Systems (NeurIPS 2024).

tumor spread in each patient. Personalizing radiotherapy to account for this spread remains an unmet clinical need in neuro-oncology [1].

The computational modeling of tumor growth has become a promising solution to this challenge [2, 3, 4, 5], by formalizing the tumor growth process through mathematical models of varying complexity [6]. However, personalizing these models to the limited clinical observational (image) data of a patient remains a formidable challenge. To address it, two main computational approaches have emerged:

**Hard-Constrained Physics-Based Models:** These models use Partial Differential Equations (PDEs) to constrain the solution space, ensuring that output tumor cell distributions adhere strictly to predefined physical laws [7, 8, 9, 10, 11].

However, since PDEs are approximations of the underlying stochastic processes in complex biological environments, they may overconstrain the system. While these models offer structure and insight, they fail to capture the intricate behaviors inherent to biological processes like tumor growth. Additionally, linking image observations to state variables requires careful consideration. As a result, the full integration of data is hindered by these limitations, reducing the effectiveness of physical models in representing empirical data.

**Data-Driven Models:** These models [12, 13] offer great flexibility by leveraging access to imaging datasets. They can directly be learned from – and applied to – the patients' diagnostic scans. However, they often lack rigorous quality control and fail to integrate crucial biological insights, such as tissue-specific infiltration patterns and brain topology. This makes them less reliable for clinical application, especially when there is insufficient data to infer them purely from imaging information.

**In Search of a Hybrid Approach:** Combining physical insights on spatiotemporal distributions of state variables with increased model flexibility could be highly beneficial. One approach to achieve this is by testing the physics residual in selected locations, as demonstrated in Physics-Informed Neural Networks (PINNs) [14, 15, 16, 17, 18]. Another way involves projecting the learned solution into a latent space that adheres to physics constraints [19, 20]. A different framework, Optimizing a Discrete Loss (ODIL) [21], constructs a residual of PDEs using grid-based discretization. Unlike methods that optimize the weights of Multilayer Perceptrons (MLPs) that implicitly parametrize the unknown fields in the PINNs method, ODIL directly optimizes the discretized unknown fields, which can be faster and more stable due to the local (rather than global, as in MLPs) dependencies of the solution on the learnable weights.

Both ODIL [22] and PINNs [23] have recently been employed to model growth equations conditioned on MRI and metabolic imaging of patients. ODIL advances state-of-the-art radiotherapy planning by relaxing the constraints of the growth equation, thereby enhancing the capture of tumor recurrence. A similar trend is observed in computer vision's novel view synthesis, where neural radiance field MLPs [24] are being replaced by discrete representations [25, 26, 27].

However, none of these hybrid approaches have modeled brain-tumor interactions dealing with constraints imposed by local structural anatomy, such as the relative composition of brain tissues and their microstructure, or by the impact of gross anatomical deformations visible in structural scans of large tumors. While biomechanical model extensions can address the latter, incorporating tissue elasticity remains challenging due to the unknown initial anatomy and the computational complexity of the tumor growth dynamics. So far, no hybrid approach free from hard PDE constraints has successfully linked biological processes, such as tumor growth, with models of the biological environment, like tissue elasticity. Only a few studies have used PINNs for biomechanical modeling in synthetic scenarios, demonstrating that the displacement field of materials can be learned from observations [28, 29].

The main contributions of this work are as follows:

- We construct discrete physics-based residuals as measures of differences between learned spatiotemporal distributions and those predicted by discretized physical equations governing tumor growth and tissue biomechanics. Using a Lagrangian perspective where particles carry tissue intensities, we project these dynamics onto static Eulerian grids to condition the learnable tumor cell and tissue distributions on available observations. Unlike data-driven models, our hybrid

method remains robust due to physics regularization while being more flexible at adapting data than hard-constrained numerical simulations.

- Our method allows for the estimation of the initial condition of the biomechanical environment. We demonstrate the ability to learn the complex unknown initial condition, which, in our case, is not just the origin of the pathology but also the initial state of the brain tissue anatomy.

- We validate our method* on the downstream task of radiotherapy planning using the largest publicly available dataset known to us, achieving new state-of-the-art performance in capturing tumor recurrence.

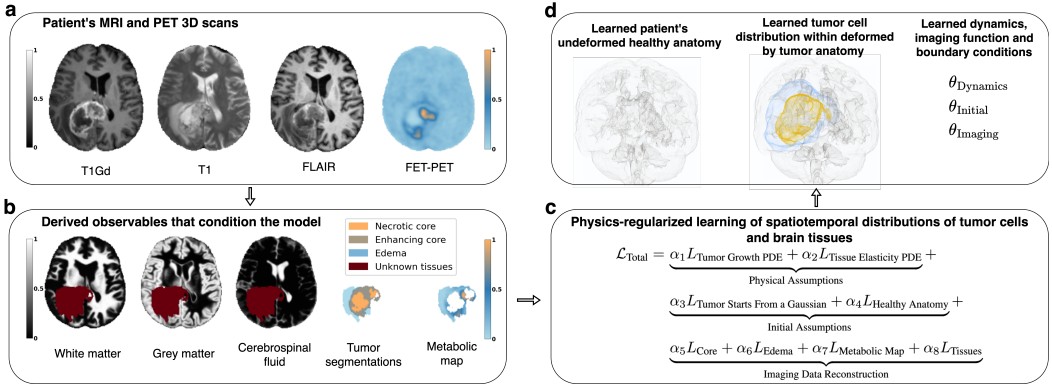

Figure 1: Method overview: (a) 3D MRI and PET scans of a glioblastoma patient (b) Preprocessed input includes brain tissue maps, tumor segmentation, and a metabolic map from FET-PET. (c) Tumor cell distribution and brain anatomy inferred using a loss function based on assumptions about physical processes, initial conditions. (d) Outcomes: initial healthy anatomy, spatial tumor cell distribution, and system identification parameters.

Figure 1 provides an overview of our method, showing how we condition the learnable full spatial tumor cell distribution on the patient data and regularize it using soft assumptions about the physics of tumor growth, tissue elasticity, and initial conditions for both the anatomy and the tumor. In addition to multiparametric MRI scans (T1Gd, T1, FLAIR), we incorporate Fluoroethyl-L-tyrosine PET (FET-PET), a specialized form of Positron Emission Tomography (PET), as an additional imaging modality that captures the metabolic activity of tumor cells.

## 2 Physical Assumptions

Our model integrates physical assumptions to ensure that the learned tumor cell distribution adheres to established biological and mechanical principles. These assumptions are incorporated into the loss function as regularization terms alongside initial condition assumptions and data terms.

### 2.1 Physical Model

We consider the problem of learning the tumor cell distribution within the domain $\Omega \times (0, 1]$, where $(0, 1]$ represents the assumed time of tumor growth, and the unit cube $\Omega = [0, 1]^3$ describes our spatial domain.

The objective is to determine a tumor cell density function $c(x, t)$, $(x, t) \in \Omega \times (0, 1]$ approximately satisfying the reaction-diffusion-advection equation [30]:

$$\frac{\partial c}{\partial t} = \underbrace{\mathcal{D}c}_{\text{cell migration through diffusion}} + \underbrace{\mathcal{S}c}_{\text{cell proliferation}} - \underbrace{\nabla \cdot (\boldsymbol{v}c)}_{\text{tissue displacements through advection}}, \tag{1}$$

where $\mathcal{D}c = \nabla \cdot (D(\boldsymbol{m})\nabla c)$ represents the diffusion of the cells with the tissue-dependent diffusion constant $D(\boldsymbol{m})$, $\mathcal{S}(c) = \rho c(1 - c)$ describes the cell growth with a growth factor $\rho$, and $\boldsymbol{v}$ is a

---

*Code is available at https://github.com/m1balcerak/PhysRegTumor

velocity field moving the cells in the interior of the domain due to the tissue's elastic properties and the tumor-induced stresses in the tissue.

The brain tissue vector $\boldsymbol{m}$, which represents the percentage-wise concentration of different tissue types, is defined as $\boldsymbol{m}(x,t) = [m_{\text{WM}}(x,t), m_{\text{GM}}(x,t), m_{\text{CSF}}(x,t)]$, where $m_{\text{WM}}$, $m_{\text{GM}}$, and $m_{\text{CSF}}$ denote the concentrations of white matter, gray matter, and cerebrospinal fluid, respectively.

The brain tissue vector $\boldsymbol{m}$ is governed by:

$$\partial_t \boldsymbol{m} + \text{div}(\boldsymbol{m} \otimes \boldsymbol{v}) = 0, \tag{2}$$

where $\otimes$ denotes the outer product. We assume Neumann boundary conditions for both $c$ and $\boldsymbol{m}$. For $c$, the no-flux condition applies at the borders of diffusive tissues, specifically the combined white and gray matter. For each tissue component in $\boldsymbol{m}$, the no-flux boundary is assumed to be the constant brain boundary visible in the MRI scans.

The tumor induced stresses lead to a displacement field $\boldsymbol{u}$, such that the velocity is given by $\boldsymbol{v} = \partial_t \boldsymbol{u}$. We assume quasi-static mechanical equilibrium due to the slow tumor growth rate relative to tissue mechanical responses, leading to:

$$\nabla \cdot \boldsymbol{\sigma}(\boldsymbol{u}) + \gamma \nabla c = 0, \tag{3}$$

where $\gamma$ regulates the impact of the tumor on the displacement and is a learnable patient-specific parameter. We apply the Neo-Hookean model [31] to represent hyperelastic behavior in biological tissues, where the stress $\boldsymbol{\sigma}$ is related to the deformation gradient $\mathbf{F}_{ij} = \delta_{ij} + \frac{\partial u_i}{\partial x_j}$ with the Lamé parameters $\lambda$ and $\mu$ (material properties in Appendix G), averaged across different tissue types to accommodate the heterogeneity:

$$\boldsymbol{\sigma} = \frac{\bar{\mu}}{J}(\mathbf{F}\mathbf{F}^T - \mathbf{I}) + \bar{\lambda}\ln(J)\mathbf{I}, \tag{4}$$

with $J = \det(\mathbf{F})$. The averaged Lamé parameters, $\bar{\lambda}$ and $\bar{\mu}$, are computed based on the proportional contributions of constituent tissues, ensuring the stress tensor accurately reflects the composite mechanical properties of the mixed tissues. This methodology facilitates detailed simulations adhering to the elasticity laws relevant to heterogeneous biological environments.

The set of learnable parameters $\theta_{\text{Dynamics}}$ describing the tumor dynamics, is thus given by $\theta_{\text{Dynamics}} = \{D_{\text{GM}}, R, \rho, \gamma\}$, where $D_{\text{GM}}$ and $D_{\text{WM}} = RD_{\text{GM}}$ are the diffusivities in pure gray and white matter regions, which are averaged to the effective diffusivity with a weighted average based on the tissue proportions.

## 2.2 Discrete Physics Residuals

We use two different approaches to discretize the elasticity equations and the reaction-diffusion-advection equations. For the former, we use a time-dependent grid in space, described by the $N_x$ points $(\boldsymbol{p}_j^n)_{1 \leq j \leq N_X}$ at time $n\Delta t$, $0 \leq n \leq N_t$, which initially form a uniform grid on $\Omega$. For the latter, we further partition the domain $\Omega$ at the $n$th point in time into $N_x$ cells $\Omega_i^n$, $1 \leq i \leq N_x$, such that $\Omega = \bigcup_{i=1}^{N_x} \Omega_i^n$. Note that the nodes defining the cell boundaries are carried by the particles $\boldsymbol{p}_j^n$, whose movement, governed by tissue elasticity, gradually deforms the cells $\Omega_i^n$ from their initial rectangular shape.

The displacement field $\boldsymbol{u}^n$ is computed based on the initial ($t = 0$) and current ($t = n\Delta t$) positions of particles $\boldsymbol{p}^n$ within the tissue by $\boldsymbol{u}^n = \boldsymbol{p}^n - \boldsymbol{p}^0$. Using central finite differences on the initial uniform reference grid, we can consecutively approximate $\nabla \boldsymbol{u}_j^n$, $\boldsymbol{F}_j^n$, $\nabla c_j^n$, $\nabla \cdot \boldsymbol{\sigma}_j^n$, and define the tissue residual:

$$L_{\text{Tissue Elasticity PDE}} = \sum_{n=0}^{N_t} \sum_{j=0}^{N_x} \left( \nabla \cdot \boldsymbol{\sigma}_j^n + \gamma \nabla c_j^n \right)^2. \tag{5}$$

As the particles, representing grid points, move with the tissue's deformation, they inherently accommodate the advection. Thus, the observed displacement of tumor cells attributed to advection is absorbed into the changes in particle positions. This leads to a simplification of the tumor equation when solved in the moving reference frame of the particles.

We use an implicit Euler scheme combined with a cell-centered finite-volume discretization to discretize the tumor equation. We denote the boundaries between two cells $1 \leq i, j \leq N_x$ by $\Gamma_{ij}^n = \Gamma_{ji}^n = \partial\Omega_i^n \cap \partial\Omega_j^n$. Further, the index set of all neighboring cells of $\Omega_i^n$ is given by $N_i = \{j : \partial\Omega_j \cap \partial\Omega_i \neq \varnothing \wedge j \neq i\}$ and assumed to stay independent of $n$, due to the extend of our deformations (see limitations in Section 7). For a function $c_i^n$ at time $n\Delta t$ at the $i$th cell, we discretize the diffusion $\mathcal{D}$ and reaction $\mathcal{S}$ operators by:

$$\mathcal{D}[c_i^n, D_i^n] = \sum_{j \in N_i} |\Gamma_{ij}| D_{ij}^n \frac{c_j^n - c_i^n}{\|\boldsymbol{x}_j^n - \boldsymbol{x}_i^n\|}, \text{ and } \mathcal{S}[c_i^n] = |\Omega_i^n| \rho c_i^n (1 - c_i^n), \tag{6}$$

where $D_{ij}^n$ is the harmonic mean of $D_i^n$ at the cell boundary, $|\Gamma_{ij}^n|$ denotes the interface area, and $|\Omega_i^n|$ denotes the cell volume. We thus obtain the tumor loss by:

$$L_{\text{Tumor Growth PDE}} = \sum_{n=1}^{N_t} \sum_{i=0}^{N_x} \left( \frac{|\Omega_i^n|}{\Delta t} (c_i^n - c_i^{n-1}) - \mathcal{D}[c_i^n, D_i^n] - \mathcal{S}[c_i^n] \right)^2. \tag{7}$$

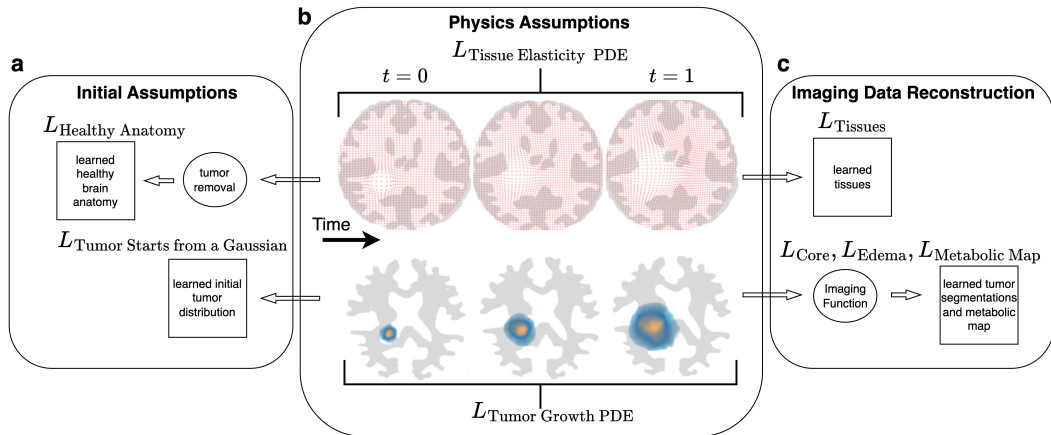

Figure 2: Learning process overview: optimization of spatiotemporal distributions of tumor cells and tissues. (a) Initial condition penalties enforce symmetric healthy anatomy, with the initial tumor distribution at $t = 0$ as a small Gaussian blob. (b) Physics penalties regularize dynamics between the initial and the final time. The first row shows gray matter contours with particle positions; the second row shows white matter contours with learned tumor concentrations. (c) Agreement of tumor distribution with anatomical tissues, visible tumor segmentations, and metabolic map after transforming the final tumor distribution through the imaging function.

## 2.3 Particle-Grid Projections

While a Lagrangian frame representation from the point of view of particles is beneficial for accurately modeling advection, an Eulerian frame is essential for rendering the advected particle states into the image space. Denoting the Eulerian and Lagrangian frames as $G$ and $P$ respectively, an Eulerian field $\mathcal{F}^G$ and a Lagrangian field $\mathcal{F}^P$ are related through:

$$\mathcal{F}_j^P \approx \sum_i w_{ij} \mathcal{F}_i^G, \quad \mathcal{F}_i^G \approx \frac{\sum_p w_{ij} \mathcal{F}_j^P}{\sum_j w_{ij}}, \tag{8}$$

where $i$ indexes grid nodes, $j$ indexes particles, and $w_{ij}$ represents the weight of the trilinear shape function defined on node $i$ and evaluated at the location of particle $j$.

## 3 Initial Assumptions

We model the initial conditions of the tumor and brain tissues by incorporating assumptions into our loss functions, ensuring the model aligns with the expected biological and anatomical state of the brain before and at the onset of tumor growth.

## 3.1 Initial Tumor Distribution

We assume that the initial tumor distribution can be represented by a small Gaussian blob. To enforce this assumption, we penalize the difference between $\mathcal{F}^G(u)^0$ and a Gaussian function centered at $x_0 \in \Omega$. The center of this Gaussian blob, $x_0$, is part of the learnable parameters $\theta_{\text{Initial}} = \{x_0, \boldsymbol{m}^0\}$. The initial condition loss for the tumor cells is defined as:

$$
L_{\text{Tumor starts from a Gaussian}} = \sum_{i=0}^{N_x} \left( \mathcal{F}^G(c)_i^0 - D_1 \exp\left( -\frac{(x_i - x_0)^2}{D_2} \right) \right)^2, \tag{9}
$$

where $D_1 = 0.5$, $D_2 = 0.02$ are constants and $x_i$ is the position of the center of $\Omega_i^0$.

## 3.2 Healthy Anatomy

The deformations are clearly visible in the MRI images and are an integral part of understanding growth dynamics. To obtain a plausible estimate of the unperturbed brain, we need to incorporate an anatomical prior.

A healthy brain is roughly symmetric between hemispheres, as justified by anatomical and functional observations. Imaging techniques such as MRI reveal symmetrical tissue structures, and electroencephalography (EEG) recordings show symmetrical patterns of electrical activity in the brain [32, 33]. This assumption applies to key tissue types, including white matter, gray matter, and cerebrospinal fluid. We construct a loss function, denoted as $L_{\text{Healthy Anatomy}}$, to quantify the asymmetry of the learned $\mathcal{F}^G(\mathbf{m}^0)$. The implementation details are provided in Appendix E. We do not use the unperturbed anatomy explicitly for inference of the tumor cells, however, we need additional constraints to bound $\gamma$ from Eq. 3 that couples tumor cells with tissue dynamics. While symmetry of the healthy brain is a coarse assumption, we soften it by quantifying the asymmetry in lower resolutions.

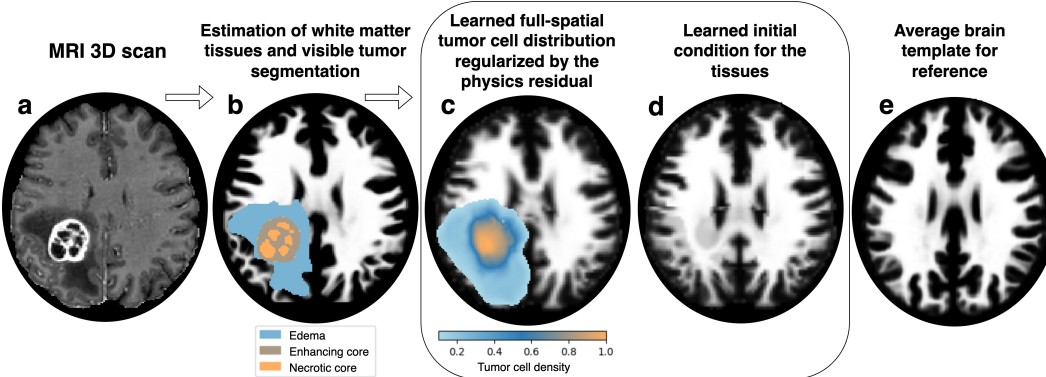

Figure 3: Inference overview: (a) Patient's MRI 3D scans. (b) Estimated tissues through non-rigid registration of the average brain, showing white matter and visible tumor segmentations. (c) Learned tumor cell distribution, regularized by the physics residual and aligned with patient data. (d) Learned initial condition of the tissues representing healthy anatomy. (e) Average brain template for reference, rigidly registered to the MRI scan. Notable anatomical differences include the lack of matter passage between the hemispheres next to the tumor, which could affect tumor cell inference results.

# 4 Imaging Data Reconstruction

Imaging data refers to information directly observed from the patient's MRI and FET-PET 3D scans [34], including visible tumor parts and surrounding brain tissues. As shown in Figure 1, this data includes estimated brain tissue probability maps, visible tumor segmentation, and a metabolic map derived from FET-PET scans, focusing on the enhancing core and edema. MRI scans provide segmentation maps [35], revealing the visible tumor. Using non-rigid atlas image registration with

tumor masking [34], we estimate tissues both outside and hidden by the tumor, including white matter (WM), gray matter (GM), and cerebrospinal fluid (CSF). These observations offer partial information about both the anatomy and the tumor cell distribution.

The imaging function (see Figure 2) [22, 36] bridges simulated tumor cell densities with visible tumor segmentation and brain tissues, as well as the metabolic map. It depends on parameters for thresholding the tumor cell distribution to obtain segmentation, denoted as $\theta_{\text{Imaging}}$, which must be learned.

We construct a loss function that minimizes the difference between model predictions and patient observables. Specifically, $L_{\text{Core}}$ matches the visible tumor core, $L_{\text{Edema}}$ matches the visible edema, and $L_{\text{Metabolic Map}}$ matches the metabolic map profile. Tissue intensities carried by particles are set to reproduce the patient's estimated tissues at $t = 1$. While these intensities are predetermined, particle positions are weakly constrained, so matching the tissues (WM, GM, CSF) to the observed structures outside the tumor is enforced through $L_{\text{Tissues}}$. The constructed penalty terms align our spatiotemporal distributions of tumor cells and tissues with patient data characteristics. Implementation details of the data losses and the imaging function are provided in Appendix D.

## 5   Combined Loss Function

The loss function $\mathcal{L}_{\text{Total}}$ is constructed to balance the contributions of physics-based regularization, assumptions about the initial conditions, and the imaging empirical data. This facilitates a comprehensive approach to infer the full spatial tumor cell distribution at the time of the patient's data acquisition (see Figure 3 for an overview of the inference process and Figure 2 for the learning process visualization):

$$
\begin{aligned}
\mathcal{L}_{\text{Total}} = & \underbrace{\alpha_1 L_{\text{Tumor Growth PDE}} + \alpha_2 L_{\text{Tissue Elasticity PDE}}}_{\text{Physical Assumptions}} + \\
& \underbrace{\alpha_3 L_{\text{Tumor Starts From a Gaussian}} + \alpha_4 L_{\text{Healthy Anatomy}}}_{\text{Initial Assumptions}} + \\
& \underbrace{\alpha_5 L_{\text{Core}} + \alpha_6 L_{\text{Edema}} + \alpha_7 L_{\text{Metabolic Map}} + \alpha_8 L_{\text{Tissues}}}_{\text{Imaging Data Reconstruction}} .
\end{aligned}
\tag{10}
$$

All $\alpha_*$ parameters were determined based on experiments with synthetic data involving both single focal and local multi-focal tumors. Extensive ablation studies are provided in Appendix C, and the details of the synthetic experiment setup can be found in Appendix F.

## 6   Validation and Ablation Study of Radiotherapy Planning

There is no objective ground truth for pre-operative tumor cell distribution, so methods are evaluated through downstream tasks like defining radiation target volumes. An "accurate" method targets areas that later show progression in follow-up MRI, acknowledging that radiation delays but does not eliminate tumor regrowth. An "inaccurate" method spares these areas or targets regions that remain tumor-free. The current "Standard Plan" defines the target volume using a 1.5 cm isosurface around the MRI-segmented tumor core, while our approach generates isosurfaces from learned tumor cell distributions. Figure 4 visualizes the Standard Plan and a plan based on our learned tumor cell distribution. Each model's target volume is simply a redistribution of the target volume of the Standard Plan.

We assess accuracy by measuring the overlap between the target volumes and regions of later tumor progression, calculating the *Recurrence Coverage* [%]. This metric represents the percentage of the tumor progression region that falls within the radiotherapy target volume of a given model, after rigid alignment of both scans.

Additional experiments studying properties of the tumor growth and imaging model on synthetic data, i.e., without relying on indirect validation via radiotherapy planning, are given in Appendix B.

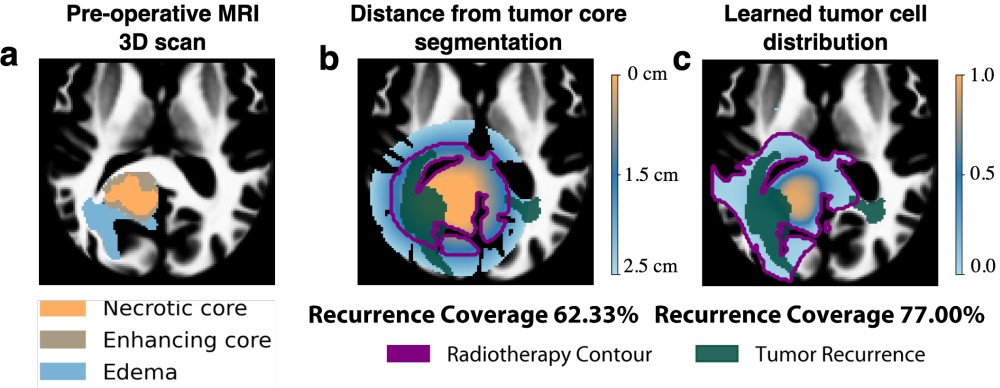

Figure 4: Radiotherapy planning: a) Pre-operative segmentations with white matter concentration as background. b) Distance map from the tumor core segmentation with a 1.5 cm contour and within diffusive tissue, constituting the Standard Plan. c) Our learned tumor cell distribution with the isosurface contour where the enclosed volume equals the total volume of the Standard Plan.

## 6.1 Implementation details

Our computational model uses a multi-resolution method [37] with additional, coarser grids to accelerate convergence. We employ four levels of grid refinement, resulting in 152,880,048 unknowns. At the finest grid level of $72 \times 72 \times 72 \times 96$, each grid point contains four unknowns: three for particle positions and one for tumor cell density.

The learning process includes system identification of unknowns: $\theta_{\text{Dynamics}}$, $\theta_{\text{Initial}}$, and $\theta_{\text{Imaging}}$, which parameterize tumor growth dynamics, tissue elasticity, initial conditions, and imaging characteristics. Using the Adam optimizer, convergence takes around 3 hours on an NVIDIA RTX A6000 GPU.

## 6.2 Dataset

We use a publicly available dataset of 58 patients with preoperative MRI scans and preoperative FET-PET imaging, along with follow-up MRI scans at the time of the first visible tumor recurrence [22].

We would like to emphasize that this is the largest publicly available dataset of this kind, i.e., with multi-modal imaging data and follow-up MRI. Its size is common for medical studies and sufficient to motivate clinical trials (see, for example, [38, 39, 40] on clinical investigations into the use of PET for radiotherapy planning). The problem we address is a critical challenge in cancer treatment research that calls for ML solutions. However, data acquisition is costly and time-consuming, and clinical datasets, such as ours with only a few dozen complete observations, are insufficient to support purely data-driven ML approaches (as visible from results reported in Table 1). At the same time, it is this inherent limitation that is encouraging us to incorporate physics-based assumptions and physics-informed regularizations into an ML model to effectively address the driving clinical problem.

The follow-up images with visible tumor recurrences are not used by our method during the optimization or hyper-parameter search at any point; therefore, the dataset can be considered 100% unseen. The decision not to use any recurrence data for hyper-parameter tuning is to avoid diluting the already small dataset.

## 6.3 Models and Features

Our analysis includes a range of models, each categorized by their distinct capabilities:

- **Numerical Physics Simulations**: Employ methods such as Finite Element Method (FEM) and Finite Difference Method (FDM) to model tumor growth dynamics and account for dynamic tissue behaviors. The dynamic parameters are determined through numerous simulations with varying parameters. The brain tissues' initial conditions are based on average brains.

- **Data-Driven Neural Networks (Unconstrained)**: Use Convolutional Neural Networks (CNNs) to directly predict likely recurrence locations, representing a population-based data-driven approach.
- **Data-Driven Neural Networks (Physics-Constrained)**: Similar to the above unconstrained approach but limited to finding parameters for physics simulations, incorporating hard physics constraints.
- **Static Grid Discretization**: A previous state-of-the-art method that uses Optimizing a Discrete Loss (ODIL) and static grids to penalize the learned spatiotemporal tumor cell distribution by quantifying the discrepancy with the tumor growth equation, keeping brain anatomy static throughout the entire process.
- **Standard Plan**: Apply uniform safety margins around the tumor core, serving as the baseline for volume determination and reflecting current clinical practice post-tumor resection surgery radiotherapy planning.

Specific implementations within these categories are explained in Appendix A.

### 6.4 Results

Table 1, specifically the "Recurrence Coverage [%] (Any)" column, and Figure 5a present results where recurrence is defined by any segmentation visible in the follow-up MRI scan. Our method achieved 74.7% recurrence coverage, surpassing the previous state-of-the-art at 72.9% and the Standard Plan at 70.0%. Using a predefined average brain reduced the performance to 73.4%. The physics-constrained data-driven approach slightly outperforms the Standard Plan at 67.1% and significantly outperforms its unconstrained variant at 59.0%. This highlights the advantage of adding physics constraints. Figure 5a shows the distance between 'Greater' and 'Less' categories increased from 26% to 36%.

Table 1, using the "Recurrence Coverage [%] (Enhancing Core)" column, and Figure 5b show results where recurrence is defined by enhancing core segmentation, per RANO guidelines [41]. Our method leads with 89.9% average recurrence coverage, compared to 89.0% for static grid discretization and 87.3% for the Standard Plan. Numerical physics simulations are more robust at 86.2% compared to 84.3% for the physics-constrained data-driven method. The unconstrained data-driven method remains the lowest at 66.8%. Results are close because the enhancing core often occurs near pre-operative MRI segmentations, leading to many methods achieving 100% coverage (seen in 'Equal' in Figure 5b). However, Figure 5b shows an increase in the preference from 19% to 28%.

In both recurrence definitions, mean estimate uncertainty is substantial, as Recurrence Coverage for individual patients can vary significantly. These uncertainties are the standard errors of the mean. Nonetheless, Figure 5 shows that even slight improvements in average Recurrence Coverage consistently enhance radiotherapy outcomes compared to the Standard Plan baseline.

Table 1: Comparison of recurrence segmentation coverage given equal radiation volume.

| Model | Recurrence Coverage[%] (Any) | Recurrence Coverage[%] (Enhancing Core) | Dynamical Tissues | Inferable Healthy Anatomy | Population-Based Data-Driven | Physics-Constrained |
|---|---|---|---|---|---|---|
| NN (Unconstrained) | $59.0 \pm 4.3$ | $66.8 \pm 4.9$ | × | × | ✓ | × |
| NN (Physics-Constrained) | $70.4 \pm 3.7$ | $84.3 \pm 3.3$ | × | × | ✓ | Hard |
| Numerical Physics Simulations | $67.1 \pm 3.8$ | $86.2 \pm 3.6$ | ✓ | × | × | Hard |
| Standard Plan | $70.0 \pm 3.8$ | $87.3 \pm 3.6$ | × | × | × | × |
| Static Grid Discretization | $72.9 \pm 3.5$ | $89.0 \pm 3.3$ | × | × | × | Soft |
| Ours (w/o inferable anatomy)* | $73.4 \pm 3.2$ | $89.3 \pm 2.9$ | ✓ | × | × | Soft |
| Ours | $\mathbf{74.7} \pm 3.1$ | $\mathbf{89.9} \pm 2.7$ | ✓ | ✓ | × | Soft |

*We set the initial condition for the brain tissues as a hard constraint, representing the average brain.

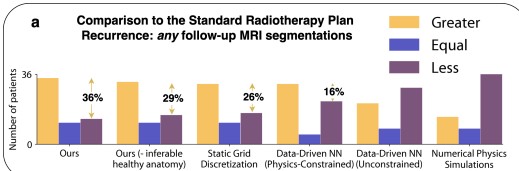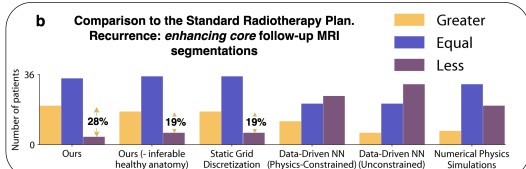

Figure 5: Direct patient-by-patient comparisons to the Standard Plan of radiotherapy plans with equal total volumes: "Greater," "Equal," and "Less" refer to the direct numerical comparison of Recurrence Coverage. a) Recurrence is defined as the union of edema, enhancing core, and necrotic core on the follow-up MRI segmentation. b) Recurrence is defined as the enhancing core on the follow-up MRI segmentation.

As shown in Figure 5, the gap between the "Greater" and "Less" categories represents the number of patients who benefit from our method compared to the standard plan. We observe a consistent improvement with our method, demonstrating superior performance beyond just higher mean recurrence coverage, which could be skewed by outliers. The results are statistically significant, as shown in Table 3 in Appendix B.

# 7    Conclusion

In this work, we present the first successful approach to a soft-constrained physics system-identification problem that combines the complex biological process of tumor growth with the biomechanical environment such as elastic brain tissues. Our method integrates physics-based constraints with multi-modal imaging data to enhance tumor treatment planning for glioblastomas. The method balances adherence to observed patient data and physics-based penalties through a unique discretization scheme, serving as a flexible spatiotemporal regularization term.

Additionally, our method provides estimates of the initial condition, i.e., tumor-free, pre-deformation anatomy. Such healthy brain anatomy can be utilized in various studies, e.g., for aligning with post-surgical scans to assess the extent of tumor removal and detect complications [42, 43, 44].

Overall, our results show that this approach outperforms previous state-of-the-art techniques in covering tumor recurrence areas, with improved performance on real-world patient data. Finally, we want to point out that relaxing physical model constraints has broad applicability beyond glioblastoma localization. This approach can extend to other real-world problems where rigid physics-based models are limited but still relevant.

**Limitations:** The convergence speed of the method depends on a multiresolution grid, assuming that problems can be represented at lower resolutions. However, this approach may not be directly applicable to PDEs with narrow-phase interfaces. Additionally, while large deformations are allowed, we assume they are not large enough to cause material rupture, which may not be optimally handled in the Lagrangian perspective without modifications to the presented scheme.

## Acknowledgments and Disclosure of Funding

We extend our gratitude to John Lowengrub and Ray Zirui Zhang from the University of California, Irvine, for their invaluable insights and discussions. This research was supported by the Helmut Horten Foundation and the DCoMEX project.

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

## A    Baselines Implementation

Here we provide details regarding the specific baseline implementations:

**Numerical Physics Simulations:**    We employ the numerical forward simulation scheme from [45] which was implemented on GPU [13]. The simulations follows the equation 1. We initialize the brain tissues with a rigidly registered average brain. For $\theta_{\text{Dynamics}}$, $x_0 \in \theta_{\text{Initial}}$, and $\theta_{\text{Imaging}}$, we use the Covariance Matrix Adaptation Evolution Strategy (CMA-ES) method [10, 46].

**Data-Driven Neural Networks (Unconstrained):**    We use nnU-Net [47] which takes brain tissue distribution, FET-PET map, and pre-operative tumor segmentations to directly predict core tumor recurrences. After predicting the recurrences, we construct a distance distribution from the predicted recurrences. This distribution is then thresholded, similar to other methods, to create a binary radiotherapy treatment map of the same volume as the Standard Plan. The network is trained using an 80%/20% train-test split.

**Data-Driven Neural Networks (Physics-Constrained):**    We employ a recently published method [13] that uses CNNs to map pre-operative tumor segmentation to $\theta_{\text{Dynamics}}$ and runs a numerical simulation following equation 1. This method uses an average brain as the initial anatomy. After inference, the tumor cell distribution is non-rigidly registered to the patient's brain anatomy.

**Static Grid Discretization:**    This method, as implemented in [22], uses estimated brain tissues (4) as the initial condition for the anatomy and later keeps the tissues static, effectively setting $\gamma \in \theta_{\text{Dynamics}}$ in Equation 3 to 0.

**Standard Plan:**    This approach uses a 1.5 cm uniform safety margin around the resection cavity and/or remaining tumor on MRI imaging, aligning with both North American and European guidelines [48, 49].

## B    Synthetic Results

To show how the method performs on tasks where there is ground truth, our method and GliODIL [16] (previous SOTA, Static Grid) are additionally evaluated on synthetic cases generated by numerical physics solvers under various conditions. See Table 2 for the results and Table 3 for statistical tests.

| | RMSE (1 Focal Origin) [%] | | RMSE (3 Focal Origins) [%] | |
|---|---|---|---|---|
| **Input Type** | Ours | Static Grid [16] | Ours | Static Grid [16] |
| Tumor Core | **10.2** $\pm$ 0.4 | 12.2 $\pm$ 0.6 | 13.3 $\pm$ 0.5 | **13.0** $\pm$ 0.7 |
| Tumor Core, Edema, and Metabolic Map | **2.5** $\pm$ 0.3 | 5.5 $\pm$ 0.8 | **4.0** $\pm$ 0.5 | 6.9 $\pm$ 0.5 |
| Tumor Core, Edema, and Metabolic Map, $\theta_{\text{Initial}}$, $\theta_{\text{Dynamics}}$ | **0.3** $\pm$ 0.1 | 3.3 $\pm$ 0.2 | **0.4** $\pm$ 0.1 | 4.2 $\pm$ 0.3 |

Table 2: Synthetic results comparing RMSE for 30 patients with different input types for one and three tumor origins, using our method and Static Grid Discretization (GliODIL [16]). The test dataset was not used for hyper-parameter tuning. Parameter ranges in Appendix D and $\gamma \in [0, 1.5]$ from Eq. 3 in the manuscript. Ours wins because GliODIL cannot deform brain tissues through its static grid limitations.

| Comparison | Recurrence (Any) | Recurrence (Enhancing Core) |
| --- | --- | --- |
| | p-value | p-value |
| Ours vs Standard Plan (1.5cm) | 0.014 | 0.00082 |
| Ours vs Static Grid Discretization (GliODIL[16]) | 0.043 | 0.039 |

Table 3: Statistical significance tests comparing recurrence coverage (any and enhancing core) among different models using the Wilcoxon signed-rank test. The data is not normally distributed which we tested using Shapiro-Wilk test resulting in p-values below 1E-12.

## C  Extended Ablation Study

Additional information regarding the loss coefficients:

$\alpha_2$ is essential for grid stability. Without it, the grid would be unstable, causing folding and rendering the results meaningless. If set to zero, $\alpha_2$ can be replaced by a Laplacian-based regularization, as visualized in Figure 6b and Figure 6c, though this clearly results in less biologically plausible deformations.

$\alpha_4$ regularizes the initial anatomy to bound $\gamma$ from Eq 3. It should be kept low due to the natural asymmetry in brains. More details are provided in Appendix E.

$\alpha_5$, $\alpha_6$, $\alpha_7$, and $\alpha_8$ are crucial for grounding PDE solutions in reality (most important, all other losses can be seen simply as regularisation of these components). Imaging data reconstruction loss cannot be calibrated using synthetic studies as all the other alphas. Instead of using recurrence data to calibrate them, we set them such that they provide a balanced contribution to the total loss

$\alpha_3$ should be kept low since the initial tumor shape is unknown. However, decreasing it too much causes a local minimum with no tumor.

$\alpha_1$ serves as the main regularization term. It should not be $\to \infty$ to avoid solutions not flexible enough to properly accommodate patient data (see Numerical Physics Simulations results in Table 1; this setup performs poorly), nor should it be $\to 0$ to prevent overfitting to imaging data alone, effectively collapsing to standard plan level performance.

Removing elements of the imaging data reconstruction loss introduces ill-posedness and degradation of synthetic results (see Table 4). Removing tumor growth physics regularizations effectively reduces our solution to performance close to the standard plan. Removing particle dynamics (static grid) reduces our model to [16] Static Grid Discretization ("GliODIL").

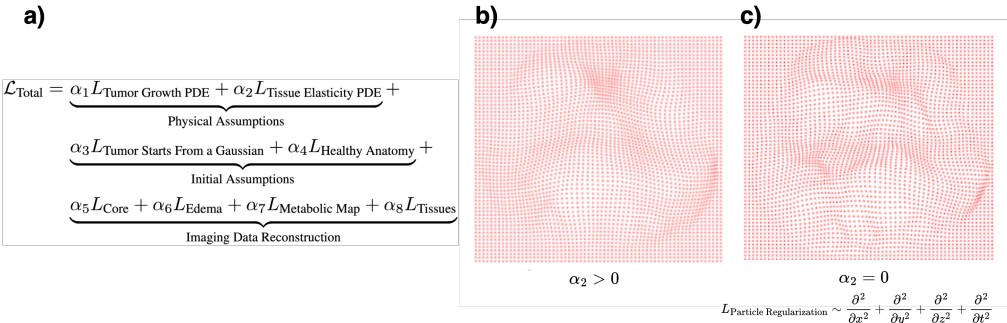

Figure 6: Deformation through particle movement: a) Total loss of our model for reference. b) Particle position with tissue elasticity. c) Particle position without tissue elasticity. These particle positions are learned for the pre-operative time. For grid stability in c), additional regularization is required. For $\alpha_2 > 0$ we observe one large localized deformation caused by the tumor, which is more biologically plausible than multiple smaller deformations visible for $\alpha_2 = 0$.

| Model Variation | Recurrence (Any) | | Recurrence (Enhancing Core) | |
|---|---|---|---|---|
| | Coverage [%] | IoU* | Coverage [%] | IoU* |
| Ours, $\alpha_5 = 0, \alpha_6 = 0, \alpha_7 = 0, \alpha_8 = 0$ | $60.5 \pm 4.9$ | $0.15 \pm 0.02$ | $83.5 \pm 3.7$ | $0.05 \pm 0.01$ |
| Ours, $\alpha_6 = 0, \alpha_7 = 0$ | $63.5 \pm 4.0$ | $0.15 \pm 0.03$ | $86.5 \pm 3.6$ | $0.06 \pm 0.02$ |
| Ours, $\alpha_3 = 0$ | $70.0 \pm 3.8$ | $0.16 \pm 0.02$ | $87.4 \pm 3.4$ | $0.06 \pm 0.02$ |
| Standard Plan (1.5cm) | $70.0 \pm 3.8$ | $0.16 \pm 0.02$ | $87.3 \pm 3.6$ | $0.04 \pm 0.01$ |
| Ours, $\alpha_1 = 0$ | $70.9 \pm 3.8$ | $0.16 \pm 0.02$ | $87.6 \pm 3.4$ | $0.05 \pm 0.02$ |
| Ours, $\alpha_2 = 0$ | $71.9 \pm 3.8$ | $0.15 \pm 0.02$ | $88.1 \pm 3.4$ | $0.05 \pm 0.02$ |
| Ours, $\alpha_4 = 0$ | $72.8 \pm 3.9$ | $0.16 \pm 0.02$ | $89.0 \pm 3.4$ | $0.06 \pm 0.02$ |
| Static Grid Discretization (GliODIL, [16]) | $72.9 \pm 3.5$ | $0.16 \pm 0.02$ | $89.0 \pm 3.3$ | $0.06 \pm 0.02$ |
| Ours | $\mathbf{74.7} \pm 3.1$ | $\mathbf{0.19} \pm 0.02$ | $\mathbf{89.9} \pm 2.7$ | $\mathbf{0.08} \pm 0.02$ |

Table 4: Ablation study showing the impact of different model variations on recurrence coverage and IoU. Large deviations account for various sizes of tumors and recurrences, yet the results remain consistent and statistically significant (Table 3). *IoU heavily penalizes the fact that the radiotherapy volumes are equal between baselines and much larger than typical recurrences. Equal volume is to make comparisons with the Standard Plan without volume/recall trade-offs.

## D   Imaging Model

The core of the method's alignment with the data is encapsulated by the loss function components, which associate the simulated outputs with key imaging traits such as the tumor core, surrounding edema, metabolic activity detected through FET-PET imaging, and visible brain tissues. This loss function depends on two learnable parameters $\theta_{\text{Imaging}} = \{\theta_{\text{down}}, \theta_{\text{up}}\}$ and is expressed as:

$$L_{\text{Imaging Data Reconstruction}} = \alpha_5 L_{\text{Core}}(c, \theta_{\text{up}}) + \alpha_6 L_{\text{Edema}}(c, \theta_{\text{down}}, \theta_{\text{up}}) + \alpha_7 L_{\text{Metabolic Map}}(c) + \alpha_8 L_{\text{Tissues}}(\boldsymbol{m}). \tag{11}$$

The loss function is composed of individual terms corresponding to distinct anatomical features:

- $L_{\text{Core}}$ relates tumor cell concentrations above the threshold $\theta_{\text{up}}$ to the tumor core region.

- $L_{\text{Edema}}$ delineates the edema area surrounding the tumor, regulated by the lower and upper thresholds $\theta_{\text{down}}$ and $\theta_{\text{up}}$.

- $L_{\text{Metabolic Map}}$ assesses the metabolic activity as indicated by FET-PET signals within the edema and core regions, using a simple correlation metric, $L_{\text{Tissues}}$ connects visible brain tissues with those inferred by the method.

These adaptive parameters $\{\theta_{\text{down}}, \theta_{\text{up}}\}$ enable the model to accommodate variations in MRI/FET-PET imaging contrasts and noise levels.

We adopt sigmoid functions to portray the gradational transitions observed at tumor region margins. The sigmoid, $\sigma(x)$, is specified as:

$$\sigma(x) = \frac{1}{1 + e^{-\beta x}}. \tag{12}$$

Here, $\beta$ modulates the steepness of the transition and is set to $\beta = 50$. For the tumor core:

$$L_{\text{Core}}(c, \theta_{\text{up}}) = \sum_{i=0}^{N_x} \sigma(\theta_{\text{up}} - c_i^{N_t} - \alpha), \tag{13}$$

for the edema:

$$L_{\text{Edema}}(c, \theta_{\text{down}}, \theta_{\text{up}}) = \sum_{i=0}^{N_x} \sigma(\theta_{\text{down}} - c_i^{N_t} - \alpha) + (1 - \sigma(\theta_{\text{up}} - c_i^{N_t} + \alpha)), \tag{14}$$

where $\alpha$ offsets the thresholds and is set to $\alpha = 0.05$.

The metabolic activity within the tumor is evaluated by the loss term $L_{\text{PET}}$, which measures the correlation between the simulated metabolic signal and actual FET-PET scan observations:

$$L_{\text{PET}}(c) = 1 - \text{corr}(c^{N_t}, p^{\text{PET}}).\tag{15}$$

Here, $\text{corr}(c^{N_t}, p^{\text{PET}})$ represents the Pearson correlation coefficient between the predicted tumor cell densities $c^{N_t}$ and the restricted FET-PET signal $p^{\text{PET}}$, across all voxels within within the edema and enhancing core regions, providing a direct measure of how well the model predictions align with the observed metabolic profiles.

## E Assumption of Brain Symmetry at Initial Time Without the Tumor

Our model employs a multi-resolution analysis to assess brain symmetry, particularly useful since the brain's symmetry is more evident at coarser resolutions. This approach is applied to key tissue types: white matter (WM), gray matter (GM), and cerebrospinal fluid (CSF), each analyzed at full, 2x, and 4x downsampled levels to accommodate both the brain's structural nuances and computational efficiency.

The process for each tissue type involves converting tissue data into particles, then calculating symmetry loss at each resolution. This calculation entails:

1. Downsample the tissue representation, using average pooling
2. Divide the tissue's downsampled representation along its height, reflecting the brain's hemispherical division.
3. Mirror one hemisphere and compare it to the other, quantifying symmetry by calculating the mean absolute difference between them.

The total symmetry loss accumulates across all tissues and resolutions to measure deviation from an ideal symmetric state:

$$L_{\text{Healthy Brain}}(\boldsymbol{m}) = \sum_{k=1}^{3} \sum_{\kappa=0}^{2} \text{calculate\_symmetry\_loss}(m_k, \text{scale\_factor} = 2^{\kappa}).\tag{16}$$

The multi-resolution approach not only addresses the limitations of assuming brain symmetry at detailed levels but also streamlines computational efforts. The smoother landscape at downsampled resolutions is especially favorable for particle movement through gradient descent.

## F Synthetic Experiments

To calibrate the loss function weights, we generated a synthetic dataset for single focal and localized multi-focal tumors by solving PDEs using a finite difference method numerical solver [13]. An average brain model was used to represent the spatial distribution of brain tissues. Tumor growth model parameters, elasticity, imaging model parameters, and 1 or 3 focal locations were varied using uniform random distributions.

The metric used was the RMSE between the learned and ground truth tumor cell distributions. In total, 100 synthetic patients were generated. Table 5 provides the details of the parameter ranges used for generating the synthetic single focal and multi-focal tumor datasets.

Table 5: Parameter ranges for generating synthetic single focal and multi-focal tumor datasets

| Shared parameters | | |
|---|---|---|
| Parameter | Min | Max |
| $D_w$ | 0.035 | 0.2 |
| $\gamma$ | 0 | 1.5 |
| $\rho$ | 0.035 | 0.2 |
| $R$ | 10 | 30 |
| $\theta_{\text{necro}}$ | 0.70 | 0.85 |
| $\theta_{\text{up}}$ | 0.45 | 0.60 |
| $\theta_{\text{down}}$ | 0.15 | 0.35 |
| $T_{\text{sim}}$ | | 100 |
| Single focal tumor center (mm) | | |
| $(x_0, y_0, z_0)$ | 57.6 | 96 |
| Multi-focal tumor centers (mm) | | |
| Tumor 1 center $(x_0^1, y_0^1, z_0^1)$ | 57.6 | 96 |
| Tumor 2 center $(x_0^2, y_0^2, z_0^2)$ | $(x_0^1, y_0^1, z_0^1) \pm 9.6$ | |
| Tumor 3 center $(x_0^3, y_0^3, z_0^3)$ | $(x_0^2, y_0^2, z_0^2) \pm 9.6$ | |

## G   Material Parameters for Biological Tissues

The table below summarizes the Young's modulus and Poisson's ratio for different tissue types used in the hyperelastic modeling of biological tissues.

Table 6: Young's modulus and Poisson's ratio for biological tissues.

| Tissue Type | Young's Modulus (Pa) | Poisson's Ratio |
|---|---|---|
| Gray Matter (GM) | 2100 | 0.4 |
| White Matter (WM) | 2100 | 0.4 |
| Cerebrospinal Fluid (CSF) | 100 | 0.1 |
| Tumor | 8000 | 0.45 |

