# OpenReview forum: "Physics-Regularized Multi-Modal Image Assimilation for Brain Tumor Localization"
_NeurIPS.cc/2024/Conference — NeurIPS 2024 poster_

### Official Review · Reviewer_S9Sg · 2024-07-11

**Soundness:** 2
**Presentation:** 3
**Contribution:** 2
**Rating:** 6
**Confidence:** 4

**Summary:**

The authors proposed a discrete physic-based loss function which can be minimized to infer the spatial distribution of tumor cell distribution, which can not be assessed by imaging data only. The loss maps imaging data to a tumor tissue biomechanical model to solve inverse problems and to better understand individual tumor progression processes. One specific promise of the model is that it can learn the unknown initial condition of the brain tissue before the pathology. Their model is validated by achieving results comparable to state-of-the-art results on the task of tumor recurrence prediction from preoperative brain MRI.

**Strengths:**

The proposed method incorporates brain tissue deformation seen on medical images to include a tissue displacement term in their biomechanical to more accurately simulate individual tumor growth. Thanks to this contribution, the model can predict an estimation of the brain tissue before the apparition of the tumor thanks to a loss term ensuring symmetry of brain tissue structures to simulate a healthy anatomy. The explanation about the different loss terms are clear and well written.

**Weaknesses:**

In the reference [16], presented as the previous state-of-the-art method for recurrence coverage prediction, it seems that they obtained better results than the results displayed in table 1. More precisely, they obtained a recurrence coverage on all tumor areas of 76.09% which is better than the method presented in the submitted paper. It appears to me that the models were validated on the same dataset, but perhaps I missed something. Could the authors clarify this point?

Following the previous point, it seems the paper lacks discussion about the use of the “Healthy anatomy” loss term. Compared to the results displayed in [16], it seems that there are no significant improvements using this new loss. How can we assert the correctness of the initial healthy brain tissue reconstruction? The authors claim that this initial state could be used to better identify post-surgical complications, but no clear method is described.

**Questions:**

- Could the authors highlight the differences between [16] and their paper? Does addition of the tissue deformation in the physical modeling really improve the results?
- Could the authors better explain the interest of the initial condition reconstruction?
- Is the data-driven neural network trained on only 20% of the 58 data from radiotherapy planning dataset? Is it enough to train such a model? Could we obtain better results by increasing the training dataset?
- How are inferred the parameters theta_up and theta_down?
- In the appendix, in section B, it seems that c_i should be written instead of u_i (equation 14).
- While it seems that visually the brain tissues exhibit symmetry, it should not be completely symmetric as we can observe on Figure 3 on the average brain template. It seems that “Healthy anatomy” term could lead to overly symmetric structures (see Figure 3 on the learned initial condition between the right and left hemisphere). It appears to me that this could be a limitation of the reconstruction method. Could the authors elaborate on this? The value of the paper could be increased with some analysis/discussion of this reconstruction.

**Limitations:**

The authors incorporated a “limitations” paragraph to address the convergence seed of their method and their tissue deformation model. However, the paper lacks some discussion in regard to their contribution compared to reference [16]. To which extent does the tissue deformation model improve the results? How can we assert the correctness of the initial condition reconstruction?

---

> ### Author Rebuttal · Authors · 2024-08-06
>
> **Thank you for your thoughtful feedback and diligent comment regarding the numbers in [16].  Please note our global response comment with additional results and explanations.**
>
> **Q1**: *Could the authors highlight the differences between [16] and their paper? Does addition of the tissue deformation in the physical modelling really improve the results?  [16] obtained a recurrence coverage on all tumor areas of 76.09% which is better than the method presented in the submitted paper*
>
> **A1**: We confirm that the original authors of [16] reported different (higher) numbers for both their method and the standard plan for the Recurrence (Any) experiment. We reproduced their numbers (76.09%) using a 2cm margin instead of the 1.5cm margin they claimed. After contacting them, they acknowledged the error and will correct it. The 2cm margin used to be standard but was later changed to 1.5cm, likely causing their mistake. Larger margins increase radiotherapy volumes, naturally raising recall for all baselines. Results for 2cm and 1.5cm cannot be directly compared. Comparing correct 1.5cm margin numbers, we outperform their method (Standard Grid Discretization, "GliODIL", cf. Table 1 in the manuscript).
>
> Our method is a generalization of [16], where authors assume $\gamma$ in our Eq 3 to be 0, which makes the tissues static throughout the growth process. This is empirically not true since the deformations are clearly visible on the MRI. While our method does not require more input data, using the deformation allows us to increase the recurrence coverage while also providing a more physically plausible solution.
>
> To the best of our knowledge, this is the first learnable approach using physics-informed losses that applies to dynamical domains with underlying physical processes on a large or any scale.
>
> **Q2**: *Could the authors better explain the interest of the initial condition reconstruction? The authors claim that this initial state could be used to better identify post-surgical complications, but no clear method is described.*
>
> **A2**: Effectively, initial condition assumptions further constrain the solution space, as just the equations alone are not enough because the equations themselves have many unknowns ( $\theta_\text{Dynamics}$ ).
>
> We need another time-point (other than preoperative) to further constrain the solution. While the initial condition in itself (position and shape of the initial tumor, initial anatomy) are not directly used to estimate the tumor cells at the preoperative timepoint, they are essential to receive a plausible unique solution. The importance of the initial reconstruction is presented in Table 1 of the one-page PDF, where setting this loss $\alpha_4$ to 0 clearly reduces performance.
>
> Besides improving tumor cell density estimation, inferring the initial brain anatomy might have further clinical uses: e.g., knowing the “healthy” anatomy prior to tumor development eases atlas-based registration approaches to perform voxel-based lesion mapping (such as in [n7]). However, such uses are beyond the scope of our evaluation.
>
> **Q3**: *Is the data-driven neural network trained on only 20% of the 58 data from radiotherapy planning dataset? Is it enough to train such a model? Could we obtain better results by increasing the training dataset?*
>
> **A3**: The split for the NN (Unconstrained) is 80/20 for train/test, meaning 80% of data is used for training and 20% for testing. We employ 5-fold cross-validation to ensure that all cases are inferred by the model without overfitting. If the 80/20 split produces very low results, it is unlikely that changing the split would significantly alter the outcome.
>
> **Q4**: *How are the parameters theta_up and theta_down inferred?*
>
> **A4**: They are inferred in the same optimization loop as the tumor cell/particle solution and dynamics, using the Adam optimizer. Their true values are practically unknown and depend on factors such as the MRI instrument's field strength, contrast, and normalization. Given the same segmentations, a large gap between theta_up and theta_down values would result in very localized, steep tumors, while a small gap would lead to diffusive tumors spreading across the brain. The knowledge about the steepness comes from the FET-PET modality, whose metabolic map signal directly correlates with the cells.
>
> **Q5**: *While it seems that visually the brain tissues exhibit symmetry, it should not be completely symmetric as we can observe on Figure 3 on the average brain template. It seems that “Healthy anatomy” term could lead to overly symmetric structures (see Figure 3 on the learned initial condition between the right and left hemisphere). It appears to me that this could be a limitation of the reconstruction method. Could the authors elaborate on this? The value of the paper could be increased with some analysis/discussion of this reconstruction.*
>
> **A5**: The symmetry of the brain hemispheres is only an approximation of reality. However, an assumption on the initial anatomy is necessary for the uniqueness of the solution (namely, the parameter $\gamma$ from Eq. 3 which couples growth of tumor cells with tissue dynamics needs at least two constraints at different time-points to be bounded). We soften the symmetry assumption by calculating it in lower resolutions (fine, downsampled 2x and 4x, cf. Appendix C). This puts more emphasis on larger structures than individual voxel tissue intensity, such as the orientation of the ventricles and rough tissue symmetries. In Fig. 3 of the main manuscript, the learned initial anatomy shows some asymmetries, i.e. in the top left and top right corners. An interesting future direction is to find even better penalties for the healthy brain assumption, such as using a neural network to identify and quantify discrepancies in reconstructed healthy brain tissues. We appreciate the suggestion to expand the discussion on this topic and will incorporate the points above in the final version.

---

> ### Author Response · Authors · 2024-08-11
>
> Dear Reviewer S9Sg,
>
> We thank you again for taking the time to read our manuscript and providing valuable feedback. We hope the rebuttal and additional experiments we provided were helpful. We hope the situation regarding [16] and the initial condition importance is now clear.
> Please let us know if there are still pressing concerns.
>
> Thank you,
> The Authors

---

> > ### Comment · Reviewer_S9Sg · 2024-08-12
> >
> > Thanks for the detailed answer.
> >
> > I updated my rating accordingly. I believe the paper is stronger with those changes.

---

> ### Author Response · Authors · 2024-08-13
>
> We appreciate the reviewer's valuable feedback and are grateful for updating the score. Please feel free to reach out if any further questions arise.

---

### Official Review · Reviewer_ZDBA · 2024-07-12

**Soundness:** 3
**Presentation:** 3
**Contribution:** 3
**Rating:** 6
**Confidence:** 4

**Summary:**

The authors propose a method using soft physics-based regularization to predict the distribution of brain tumor. The main contribution is a novel discretization scheme of physics equations to model brain tumor.

**Strengths:**

1. This work is an improvement for physics-based approaches for medical imaging

2. The proposed method of discretizing the physics residuals is novel and potentially applicable to related problems and of interest to the research community

3. The paper is clearly written and the technical details are sound and sufficient

**Weaknesses:**

1. The main weakness is the lack of ground truth data to train and evaluate the model performance. The evaluation is performed not on the original task of brain tumor detection but rather on a downstream task of “recurrence coverage”, which measures the percentage of the tumor detected in follow-up MRIs (rather than the original).

2. Related to above, the authors only use one metric to evaluate their model, which may be insufficient, especially since there is no ground truth labels

**Questions:**

1. Is there any bias in the evaluation metrics since there are no ground truth labels and authors use recurrence coverage?

**Limitations:**

1. The authors should discuss the limitations of their methods in more details, especially related to the lack of ground truth labels to train the model

---

> ### Author Rebuttal · Authors · 2024-08-06
>
> **Thank you for your thoughtful feedback regarding the ground truth labels and limited amount of available data. Please note our global response comment with additional results and explanations. Below we address your specific questions.**
>
> **Weakness**: *Lack of ground truth data to train and evaluate the model performance. The evaluation is performed not on the original task of brain tumor detection but rather on a downstream task.*
>
> **Answer**: We acknowledge that this aspect of the problem was not adequately communicated in the manuscript. We thank the reviewer for raising this issue, which has enabled us to refine our message in the discussion.
>
> Personalizing radiotherapy is a critical but currently unmet need in neuro-oncology. This problem is clearly complicated by the fact that there is indeed no ground truth for the spread of tumor cells into the brain. To solve this problem, we developed a physics-informed framework that estimates this tumor cell spread from pre-operative imaging, with the clear aim of informing personalized radiotherapy for glioblastoma patients despite the lack of reliable, vast amounts of training data for purely data-driven approaches. To evaluate the quality of our predictions and compare it to several baselines, we consequently evaluated all models against the relevant clinical task for which they were developed. Estimating the recurrence coverage and comparing it to standard clinical radiotherapy plans is a well-established metric in this field [n4,n5]. We, therefore, disagree with the notion of this being a weakness of our work, but rather want to emphasize that our method - which is the first physics-informed framework that tackles deformations and the underlying biological processes in any field - excels at solving the highly challenging and clinically relevant task of personalized radiotherapy planning. We outperform all evaluated methods with statistically significant results (Table 3, one-page global PDF).
>
> **Question**: *Is there any bias in the evaluation metrics since there are no ground truth labels and authors use recurrence coverage?*
>
> **Answer**: We now understand that communicating results only through a domain specific metric is limiting our message. In addition, there are scenarios where major parts of our method could be applicable to problems that have ground truth labels/distributions. Following the reviewer's suggestion, our method and the previous state-of-the-art [16] have been additionally evaluated on synthetic cases generated by numerical physics solvers under various conditions (both ill-posed and well-posed, violating modeling assumptions about the initial condition of the tumor and adhering to them). This is highlighted in Table 2 of the attached one-page PDF.
>
> The Recurrence coverage metric we employed is a clinically relevant metric that directly evaluates the models’ performance on the clinical task they were developed for. By aiming to encompass areas of high tumor cell density (which are not visible on standard MRI, but are prone to give rise to recurrence during the disease course) into the radiotherapy plan while keeping the total radiation volume constant with respect to the standard plan, recurrence coverage immediately quantifies any improvements in radiotherapy planning. However, we agree that it is important to judge models through multiple metrics’ lenses and have, therefore, now added also IoU alongside the recurrence coverage.

---

> ### Comment · Reviewer_ZDBA · 2024-08-08
>
> I thank the authors for carefully addressing the reviewers' comments and improving the paper accordingly. I do not have further questions or suggestions on the paper.
>
> I updated my recommendation from "borderline accept" to "weak accept" as I believe the paper is stronger after the revisions.

---

> > ### Author Response · Authors · 2024-08-11
> >
> > We appreciate the reviewer's valuable feedback and are grateful for raising the score from 5 to 6. Please feel free to reach out if any further questions arise.

---

### Official Review · Reviewer_YNED · 2024-07-12

**Soundness:** 3
**Presentation:** 2
**Contribution:** 2
**Rating:** 4
**Confidence:** 4

**Summary:**

This paper presents a new approach to integrate physics-based tumor growth constraint with multi-modal imaging data to predict tumor cell distribution and thus enhance tumor treatment planning for glioblastomas. The core of the methods includes a discrete physics residual and initial assumptions encoding initial tumor distribution and symmetrical pattern between hemispheres. Experiments were performed on a small real dataset, and the success was measured by the “recurrence coverage” defined as the percentage of tumor (based on follow-up MRI) within the model-predicted volume. Experimental results demonstrate improved recurrence coverage of the presented method compared to fully data-driven or fully physics-based methods.

**Strengths:**

Ability to integrate physics-based constraints and multi-modal imaging data is important and a difficult problem to address. The presented work is thus of important potential in bringing physics-informed learning into biomedical tasks.

The dataset and task considered are quite complex involving multi-modal images and pre-operative and follow-up comparisons. I applaud the authors for the efforts devoted in this type of challenging tasks.

**Weaknesses:**

The experimental evaluation of the presented work was performed on a very small dataset with unclear training-test split. This raises some question about the general conclusion that can be drawn from the results obtained on limited samples.

Given the reported standard deviation (eg. Table 1), the obtained margin of improvement appears to be marginal without statistical significance.

The writing of the paper lacks clarify at places. For instance, the results presented in Figure 5 was unclear. What are the “greater”, “less”, and “equal” categories, and why a bigger gap between the “greater” and “less” categories indicate a favorable performance?

The objective of the presented model includes a large number of terms. The sensitivity of the model performance to the include/exclusion of these different terms and their hyperparameters deserves substantial analyses that are missing in the current paper.

**Questions:**

Clarifications on the data split used to derive the reported results will be appreciated.

The statistical significance of the reported performance (Table 1) will be appreciated.

Clarifications on the results related to Fig 5 (see questions above) will be appreciated.

How sensitive are the presented methods to the different terms in the loss functions, and which are the most important terms?

How is asymmetry measured as an optimization objective?

**Limitations:**

The authors presented some discussion about the limitations of the current work. Adding discussion about limitations related to the limited sample size for the experiments would be appreciated.

---

> ### Author Rebuttal · Authors · 2024-08-06
>
> **Thank you for your thoughtful feedback regarding the dataset size and lack of statistical tests, despite relatively high uncertainties. We appreciate acknowledging the challenges of the task.  Please note our global response comment with additional results and explanations.**
>
> **Q1:** *Clarifications on the data split used to derive the reported results will be appreciated. Additional weakness:  performed on a very small dataset.*
>
> **A1:** Please let us mention this is the largest publicly available dataset of this kind, including metabolic PET imaging alongside MRI, and follow-up with recurrence. For example, in [n1], [n2], and [n3] PET-based radiotherapy planning has been evaluated in comparably sized datasets. However, from a machine learning perspective, the size of the dataset is what effectively forces us to use physics-based assumptions and regularization.
> Our hyperparameters were selected in synthetic studies with the methodology discussed in Appendix D. Our method does not use recurrence images at any stage other than for final testing, and the synthetic patients have fixed seeds for reproducibility purposes. The clinical dataset is, therefore, a fully unseen test set for our results. Only the NN (Unconstrained) baseline was trained on recurrences since it is purely data-driven and with 80/20 train/test split over 5x cross-folding, and we report only the results from the unseen folds.
>
> **Q2:** *The statistical significance of the reported performance (Table 1) will be appreciated.*
>
> **A2:** Please see Table 3 in the one-page PDF with the tests. Our results are statistically significant and will be included in the final manuscript thanks to the reviewer suggestion. We understand the skepticism drawn from Table 1 in the main manuscript alone.
>
> **Q3:** *Clarifications on the results related to Fig 5 (see questions above) will be appreciated.*
>
> **A3:** We appreciate your comment and recognize that our explanation could have been clearer. In Figure 5, “Greater,” “Less,” and “Equal” refer to the quality of patient outcomes given radiotherapy from various methods (denoted below each histogram) compared to the standard plan in terms of recurrence coverage. “Greater” quantifies how many patients would have received a better plan, “Less” - how many would have received a worse plan. The gap between “Greater” and “Less” shows how many patients would benefit from our method compared to the standard plan. We observe a consistent improvement with our method, illustrating superior performance beyond just higher mean recurrence coverage, which could be skewed by outliers. This was meant to complement Table 1, but in hindsight, statistical test results with low p-values would be more straightforward (as in Table 3 of the one-page PDF).
>
> **Q4:** *How sensitive are the presented methods to the different terms in the loss functions, and which are the most important terms?*
>
> **A4:** We agree that an extension of the ablation studies is valuable, and we have provided such studies in Figure 1 and Table 1 of the one-page PDF. Additional information regarding the loss coefficients (see Figure 1a of the one-page PDF for reference regarding alphas):
>
> $\alpha_2$ is essential for grid stability. Without it, the grid would be unstable, causing folding and rendering the results meaningless. If set to zero, $\alpha_2$ can be replaced by a Laplacian-based regularization, as visualized in Figure 1b and 1c, though this clearly results in less biologically plausible deformations.
>
> $\alpha_4$ regularizes the initial anatomy to bound $\gamma$ from Eq 3. It should be kept low due to the natural asymmetry in brains. More details are provided in Appendix C.
>
> $\alpha_6$, $\alpha_7$, $\alpha_8$, and $\alpha_9$ are crucial for grounding PDE solutions in reality (most important, all other losses can be seen simply as regularisation of these components). Visible data loss cannot be calibrated using synthetic studies as all the other alphas. Instead of using recurrence data to calibrate them, we set them such that they provide a balanced contribution to the total loss
>
>  $\alpha_3$ should be kept low since the initial tumor shape is unknown. However, decreasing it too much causes a local minimum with no tumor.
>
> $\alpha_1$ serves as the main regularization term. It should not be $\rightarrow \infty$ to avoid solutions not flexible enough to properly accommodate patient data (see Numerical Physics Simulations results in Table 1 of the manuscript; this setup performs poorly), nor should it be $\rightarrow 0$ to prevent overfitting to visible data alone, effectively collapsing to standard plan level performance.
>
> Removing elements of the visible data loss introduces ill-posedness and degradation of synthetic results (see Table 2 of the one-page PDF). Removing tumor growth physics regularizations effectively reduces our solution to performance close to the standard plan. Removing particle dynamics (static grid) reduces our model to [16] Static Grid Discretization (“GliODIL”).
>
> **Q5:** *How is asymmetry measured as an optimization objective?*
>
> **A5:** Asymmetry is measured by calculating the total symmetry loss across different tissue types (white matter, gray matter, cerebrospinal fluid) and resolutions (full, 2x, 4x downsampled).
>
> The steps are:
> * Downsample the tissue data.
> * Divide the downsampled data into two halves, corresponding to the brain's hemispheres.
> * Mirror one hemisphere and compare it to the other by calculating the mean absolute difference.
>
> Please see Appendix C for a formal definition of the symmetry ('Healthy Brain') loss.

---

> ### Author Response · Authors · 2024-08-11
>
> Dear Reviewer YNED,
>
> We thank you again for taking the time to read our manuscript and providing valuable feedback.
> We hope our rebuttal and the additional experiments were helpful in clarifying the data splits, demonstrating the statistical significance of our findings, and highlighting the importance of each loss term.
> Please let us know if there are still pressing concerns.
>
> Thank you,
> The Authors

---

> > ### Comment · Reviewer_YNED · 2024-08-12
> >
> > I'd like to thank the authors for the effort put in the rebuttals, which has helped addressed some of my previous concerns especially in adding significance test to the experimental results. I do acknowledge the value of the study especially the use of physiological terms to enable optimization given small data. I do however believe that the sensitivity of the method performance to the various hyperparameters used in the loss function deserves a much more comprehensive study -- the rebuttal did a good job demonstrating how the absence/presence of these individual terms would affect the model performance, as well as their rough range of values (large or small). What need to be demonstrate however are the best practice for tuning these individual terms, and how their values affect the margins of improvement the method is currently demonstrating, i.e, how sensitive is the demonstrate margin of improvement to the tuning of the many hyperparameters in the loss, and what would be the strategy to obtain such optimal hyperparameters in real world settings when given a new dataset. I did raise my rating although I think more work is needed to clarify this issue considering the many hyperparameters and their effect on the margin of improvements demonstrated.

---

> ### Author Response · Authors · 2024-08-13
> **HP search, a general scheme**
>
> Thank you for your valuable feedback and for updating the score. The balance between the importance of PDEs and data is indeed a broader challenge in the physics-informed ML field. The reviewer rightly highlights a crucial issue in this area. For such an ML problem, it is essential to use a diverse set of complementary assumptions and regularizations because, without them, the problem becomes ill-posed. All the terms we included are well-grounded with clear physical motivation.
>
> Regarding sensitivity, the solution should not be overly sensitive to individual changes. We refer to Fig. 2a (and to some extent, Fig. 2e) in [16] (Static Grid Discretization method), where the authors observed a 20% difference in DICE scores by varying $\lambda_{PDE}$ across five orders of magnitude (which corresponds to $\alpha_1$, in our case), which can be seen as low sensitivity to small changes.
>
> When it comes to actual values for hyperparameter selection, in our work, we opted for a general approach using calibration on synthetic data. We began by determining the order of magnitude for the relevant hyperparameters (e.g., ensuring no tissue folding and preventing the initial tumor cells from vanishing) through a grid search, followed by further fine-tuning using the same grid search method and the RMSE as the metric (as in Table 2 of the one-page PDF). While we do not claim that the parameters we identified are necessarily optimal, they are effective enough to outperform all other baselines without raising concerns about an overly complex hyperparameter optimization scheme that might not generalize to other datasets. For datasets involving pre-operative tumor images, our method should work out-of-the-box, as it relies solely on adherence to the pre-determined growth equations and tissue elasticity, without needing specific information from our dataset. We will extend Appendix D to provide more detailed information on the hyperparameter selection procedure involving grid search and the individual physical motivations behind each parameter to assist researchers in applying our framework to new domains/equations. We are always happy to consult further on this.
>
> As a final point, we would like to emphasize that almost all physics-informed ML frameworks rely on implicitly optimizing the unknown fields using CNNs or MLPs, which introduces numerous additional hyperparameters for these architectures. In contrast, our approach only requires setting the resolution aligned with the data's resolution and the stability conditions of the corresponding growth PDEs, thereby eliminating the need to include these parameters in the hyperparameter search.

---

### Official Review · Reviewer_smuQ · 2024-07-13

**Soundness:** 3
**Presentation:** 4
**Contribution:** 3
**Rating:** 5
**Confidence:** 1

**Summary:**

This paper presents a method for brain tumor modeling through a joint data-driven and physics-based approach. The proposed approach leverages a multi-task loss that incorporates physical assumptions and prior healthy anatomy to more faithfully model the tumor cell distribution. This method outperforms existing computational approaches and the clinical standard with respect to recurrence coverage

**Strengths:**

- The presentation quality is extremely high. Care was taken to logically organize sections of the paper with helpful illustrations and definitions throughout.
- The treatment of related work is thorough, and domain-specific details are effectively communicated throughout.
- Each component of the proposed method is well-motivated and specifically catered to the application domain.
- Experiments appear to be sound, and the proposed method outperforms relevant baselines.

**Weaknesses:**

- I am slightly confused by the recurrence coverage metric. Has this metric been used in prior work? Why not a well-established overlap metric such as intersection over union, for example?

**Questions:**

- According to line 207, recurrence coverage is the “percentage of the tumor segmentation… that is encompassed within the radiotherapy target volume defined by each model.” If this is true, then wouldn’t a predicted radiotherapy target volume consisting of the entire scan obtain 100% recurrence coverage? Unless I am misunderstanding, an existing overlap metric (which penalizes such “overprediction”) might be more appropriate.
- Line 5: “Deep-learning based” -> “Deep learning-based”
- Line 234: Missing parenthetical “(Physics-Constrained:”

**Limitations:**

Limitations are briefly addressed in Section 7.

---

> ### Author Rebuttal · Authors · 2024-08-06
>
> **Thank you for your thoughtful feedback regarding the recurrence coverage. Please note our global response comment with additional results and explanations. Below, we address your specific questions.**
>
>
> **Q1**: *According to line 207, recurrence coverage is the “percentage of the tumor segmentation… that is encompassed within the radiotherapy target volume defined by each model.” If this is true, then wouldn’t a predicted radiotherapy target volume consisting of the entire scan obtain 100% recurrence coverage? Unless I am misunderstanding, an existing overlap metric (which penalises such “overprediction”) might be more appropriate.* **Weakness**: *Why not a well-established overlap metric such as intersection over union, for example?*
>
> **A1**: We apologize for not communicating it properly in the manuscript. The radiotherapy target volume is fixed and equal to the standard plan total volume. The standard plan was generated according to the current clinical guidelines [n6] (1.5cm around the visible preoperative tumor). The radiation volume is bounded by the standard plan volume and is taken equal among baselines, typically ranging from 10% to 25% of the brain volume depending on the extent of the preoperative tumor.
> Both metrics (IoU and recurrence coverage) reward overlap and penalize overprediction but in different ways. Recurrence coverage can be seen as recall and is calculated as $\frac{|R \cap V|}{|R|}$, where R is the set of recurrence voxels and V is the model-selected radiotherapy volume. The volume |V| is always equal across all baselines for a given patient, defined by the total volume of the standard plan. IoU heavily penalizes a solution for being larger than the eventual recurrence. The small IoU values across all baselines may be misleading without the context, as well as biased towards a few large recurrences (|R| comparable with |V|), skewing the metric. However, we absolutely agree that reporting more metrics might help to contextualize our results. Therefore, based on the reviewer's suggestion, we have included the IoU in Table 1 of the one-page PDF as well as the RMSE for synthetic results in Table 2 of the same document.
>
> In addition, it would not be easy to compare models with different recalls and different volumes, as it is unclear how to handle the trade-offs in the clinical setup.
>
> **Q2,Q3**: *Line 5: “Deep-learning based” -> “Deep learning-based”, Line 234: Missing parenthetical “(Physics-Constrained:”.*
>
> **A2,A3**: Thank you for pointing this out.

---

> > ### Comment · Reviewer_smuQ · 2024-08-07
> >
> > I acknowledge that I have read the authors' rebuttal and appreciate their hard work. In particular, thank you for clarifying the recurrence coverage metric and including other overlap metrics.

---

> ### Author Response · Authors · 2024-08-11
>
> We want to thank the reviewer again for the valuable suggestions.
> If we have now addressed all your concerns and questions, we would appreciate if you could consider updating your scores to reflect your post-rebuttal opinion.
> Please kindly let us know if you have any follow-up questions or areas needing further clarification.

---

### Author Rebuttal · Authors · 2024-08-06

Dear Reviewers,

**Please see the one-page PDF with additional ablation studies, synthetic data results (with ground truth), and statistical tests for the recurrence coverage results.**

We thank all reviewers for their constructive feedback that helped us improve the paper. We are encouraged by their comments:

* "I applaud the authors for the efforts devoted to this type of challenging task."

* "The proposed method of discretizing the physics residuals is novel and potentially applicable to related problems and of interest to the research community."

* "The explanation about the different loss terms is clear and well written."

* "The presentation quality is extremely high."

We have been working diligently to address your critiques: specifically, the use of a single metric in our experiments, the absence of an extended ablation study, and resolving some communication issues related to the presentation of our results, the train/test split, and addressing the size of our dataset compared to other domains.

**Train/test split of our model, size of the dataset**

We would like to emphasize that this is the largest publicly available dataset of this kind (with the follow-up scans). This size of data is standard for medical studies and sufficient to motivate clinical trials (see, for example, [n1,n2,n3] on clinical investigations into the use of PET for radiotherapy planning). The problem we address is an example of a vital (cancer research) problem with very costly and time-consuming data gathering but because it is so important to the community and to improve patient outcomes, researchers need to develop new ML methods with these limitations in mind. We have now added this explanation to the discussion.
The size of the dataset of 58 patients is relatively small compared to other ML domains, which effectively forces to use physics-based assumptions and physics-informed regularizations to approach the problem, which is not solvable otherwise (as shown by the weak results of purely data-driven methods in Table 1 of the manuscript). Despite the limited data size, our improvements are statistically significant (see Table 3 in the one-page PDF).

The follow-up images with visible tumor recurrences are not used by our method during the optimization or hyper-parameter search at any point; therefore, the dataset can be considered 100% unseen. The decision not to use any recurrence data for hyper-parameter tuning is to avoid diluting the already small dataset.

**Ground truth labels and generalization**

Personalizing radiotherapy is a challenging task because there is no ground truth for the true spread of tumor cells into the brain. To meet this urgent clinical need, we developed our framework precisely to handle this scenario. Thus, our framework was tuned using well-defined synthetic results in various scenarios (Appendix D) and only validated in the clinical situation of radiotherapy planning against the later recurrence. We would like to highlight that our framework is the first physics-informed framework that estimates deformations and underlying biological processes successfully on a large or any scale.

To show how the method performs on tasks where there is ground truth, thanks to the reviewer ZDBA suggestions, our method and the previous SOTA are additionally evaluated on synthetic cases generated by numerical physics solvers under various conditions (cf. Tab.2 in the PDF and extended discussion in the answers to ZDBA).

**Recurrence Coverage metric**

Recurrence coverage is an established metric [n4, n5], motivated by clinical translation. Given the potential side effects of radiotherapy, it is crucial to balance recurrence coverage against radiotherapy volume. By keeping the volumes constant, we eliminate the need to consider trade-offs related to volume. Recurrence Coverage is defined as recall $\frac{|R \cap V|}{|R|}$ where $R$ is the set of recurrence voxels and $V$ is the predicted radiotherapy volume. Importantly, $|V|$ is always equal across all baselines for a given patient and is defined by the total volume of the standard plan (1.5cm volume around the preoperative tumor core).
In the PDF, we have included (thanks to the suggestion from reviewer smuQ) results for IoU: $\frac{|R \cap V|}{|R \cup V|}$. The values are relatively small across all methods since $|R| \ll |V|$, leading to $|R \cap V| \ll |R \cup V|$, but can bring new perspective.

**Parameter importance**

The tumor cell localization task requires many assumptions to constrain the problem. Please refer to Figure 1a in the one-page PDF for information on the different loss terms (expanded in answers to reviewer YNED). Extensive ablation experiments are presented in Table 1 of the PDF,  thanks to the reviewer YNED suggestion.

**Symmetric healthy brain assumption**

We do not necessarily need to infer the initial anatomy for inference of the tumor cells, however, we need additional constraints to bound $\gamma$ from Eq. 3 that couples tumor cells with tissue dynamics. While symmetry of the healthy brain is a coarse assumption (rightly pointed out by reviewer S9Sg), we soften it by quantifying the asymmetry in lower resolutions (see Appendix C). Previous SOTA [16] assumes static tissues, which is an even coarser assumption, since the deformations are clearly visible in the MRI images.

[n1] Rieken et al., Radiother. Oncol., 2013. doi:10.1016/j.radonc.2013.06.043

[n2] Munck Af Rosenschold et al., Neuro-Oncol., 2015. doi:10.1093/neuonc/nou316

[n3] Grosu et al., Int. J. Radiat. Oncol. Biol. Phys., 2005. doi:10.1016/j.ijrobp.2005.02.059

[n4] Lipkova et al., IEEE Trans. Med. Imaging, 2019. doi:10.1109/TMI.2019.2902044

[n5] Metz et al., Neuro-Oncol. Adv., 2024. doi:10.1093/noajnl/vdad171

[n6] Niyazi M, et al., Radiother Oncol. 2023;184:109663. doi: 10.1016/j.radonc.2023.109663

[n7] Lai A, et al., J Clin Oncol. 2011;29(34):4482-90. doi: 10.1200/JCO.2010.33.8715

---

### Decision · Program_Chairs · 2024-09-25

**Decision:**

Accept (poster)

**Comment:**

This work presents a new approach to integrate physics-based tumor growth constraint with multi-modal imaging data to predict tumor cell distribution and thus enhance tumor treatment planning for glioblastomas. The core of the methods includes a discrete physics residual and initial assumptions encoding initial tumor distribution and symmetrical pattern between hemispheres. Experiments were performed on a small real dataset, and the success was measured by the recurrence coverage defined as the percentage of tumor (based on follow-up MRI) within the model-predicted volume. Experimental results demonstrate improved recurrence coverage of the presented method compared to fully data-driven or fully physics-based methods.

The presented work is of important potential in bringing physics-informed learning into biomedical tasks and demonstrates the ability to integrate physics-based constraints and multi-modal imaging data. The proposed method of discretizing the physics residuals is novel and potentially applicable to related problems and of interest to the research community. The paper is clearly written and the technical details are sound and sufficient.

The reviewers have some concerns: the experimental evaluation of the presented work was performed on a very small dataset with unclear training-test split. This raises some question about the general conclusion that can be drawn from the results obtained on limited samples. Given the reported standard deviation, the obtained margin of improvement appears to be marginal without statistical significance. The objective of the presented model includes a large number of terms. The sensitivity of the model performance to the include/exclusion of these different terms and their hyperparameters deserves substantial analyses that are missing in the current paper. Further concerns are about the evaluation: the main weakness is the lack of ground truth data to train and evaluate the model performance. The evaluation is performed not on the original task of brain tumor detection but rather on a downstream task of “recurrence coverage”, which measures the percentage of the tumor detected in follow-up MRIs (rather than the original). Related to above, the authors only use one metric to evaluate their model, which may be insufficient, especially since there is no ground truth labels.

The rebuttal addressed these concerns, and the paper has been improve accordingly. Most reviewers are convinced by the rebuttal.  The work addresses an important research problem and shows great promise, it has the potential to be highly impactful.